# Design and Analysis of Artificial Neural Network (ANN) Models for Achieving Self-Sustainability in Sanitation

**Mahesh Ganesapillai** [1,*] , **Aritro Sinha** [1] , **Rishabh Mehta** [1] , **Aditya Tiwari** [1] , **Vijayalakshmi Chellappa** [2] **and Jakub Drewnowski** [3,*]

1 Mass Transfer Group, School of Chemical Engineering, Vellore Institute of Technology, Vellore 632014, India; aritro.sinha2019@vitstudent.ac.in (A.S.); rishabh.mehta2019@vitstudent.ac.in (R.M.); aditya.tiwari2019@vitstudent.ac.in (A.T.)

2 Department of Statistics and Applied Mathematics, Central University of Tamil Nadu, Thiruvarur 610005, India; vijayalakshmi@cutn.ac.in

3 Department of Sanitary Engineering, Faculty of Civil and Environmental Engineering, Gdansk University of Technology, ul. Narutowicza 11/12, 80-233 Gdansk, Poland

* Correspondence: maheshgpillai@vit.ac.in (M.G.); jdrewnow@pg.edu.pl (J.D.); Tel.: +91-(97)-9029-9447 (M.G.); +48-501-516-765 (J.D.)

**Abstract:** The present study investigates the potential of using fecal ash as an adsorbent and demonstrates a self-sustaining, optimized approach for urea recovery from wastewater streams. Fecal ash was prepared by heating synthetic feces to 500 °C and then processing it as an adsorbent for urea adsorption from synthetic urine. Since this adsorption approach based on fecal ash is a promising alternative for wastewater treatment, it increases the process' self- sustainability. Adsorption experiments with varying fecal ash loadings, initial urine concentrations, and adsorption temperatures were conducted, and the acquired data were applied to determine the adsorption kinetics. These three process parameters and their interactions served as the input vectors for the artificial neural network model, with the percentage urea adsorption onto fecal ash serving as the output. The Levenberg–Marquardt (TRAINLM) and Bayesian regularization (TRAINBR) techniques with mean square error (MSE) were trained and tested for predicting percentage adsorption. TRAINBR was demonstrated in our study to be an ideal match for improving urea adsorption, with an accuracy of R = 0.9982 and a convergence time of seven seconds. The ideal conditions for maximum urea adsorption were determined to be a high starting concentration of 13.5 g.L$^{-1}$; a low temperature of 30 °C, and a loading of 1.0 g of adsorbent. For urea, the improved settings resulted in maximum adsorption of 92.8%.

**Keywords:** optimization algorithm; closed-loop sanitation; human waste management; Bayesian regularization; Levenberg–Marquardt algorithm; adsorption





## 1. Introduction

Inadequate sanitation and a lack of personal hygiene exacerbate the spread of infectious diseases such as typhoid, cholera, diarrhea, and amoebic dysentery [1–3]. These water-borne diseases affect one-quarter of the world's population under five [4]. An acute lack of sanitation infrastructure establishes a behavioral tendency toward open defecation, which increases the probability of intestinal worms, diarrhea, and environmental enteric dysfunctions [5]. According to UN estimates, over 5% of the world's population still defecates openly due to a lack of access to basic sanitation facilities, with the majority of this population concentrated in two regions—233 million people in Southern and Central Asia and another 197 million in Sub-Saharan Africa [6]. This situation is all the more disadvantageous for developing countries with high populations such as India. The situation deteriorates further as the population density is high due to increased exposure to fecal pathogens.

On the other hand, the UN University's Centre for Water Environment and Health predicts that if all human excrement from latrines were collected and burned above 300 °C, up to 8.5 million tonnes of fecal ash might be generated. Additionally, this will assist in extracting nutrients from the soil and provide a source of revenue. Similarly, wastewater streams are nutrient-dense and include plant leftovers used in various agricultural applications [7,8]. This shows unequivocally that human feces offer a unique perspective into the issues of emerging contaminants. However, increasing the number of toilets is not enough to combat water-borne diseases or achieve the Sustainable Development Goals. Improved methods of collection and treatment of human feces also play a detrimental role in creating a safe and efficient sanitation system [9,10].

Fecal sludge contains several nutrients that may be used in various applications, and hence, there is a rise in demand for the development of treatment methods for the same [3]. Many technologies have been developed for decontaminating wastewater streams, such as ion exchange, adsorption, activated sludge, membrane separation, and advanced oxidation [11]. Adsorption is a process of sequestrating contaminants by an adsorbent, typically a highly porous substance with excellent selectivity. It is primarily produced from wastes generated from industries, agricultural fields, domestic/municipal garbage [12–15]. Adsorption is also favored because it is low cost, simple to scale up, efficient on both small and big scales, and uses less energy, making it a step toward carbon neutrality. Although several isotherm models may be employed to describe adsorption processes, the Langmuir (single-layer homogeneity) and Freundlich (multilayer heterogeneity) isotherms are two of the most common ones used [16].

With artificial neural networks (ANN) emergence, a new model for adsorption has gained prominence [17–19]. Artificial neural network (ANN) is a computing technique associated with artificial intelligence, drawing similarities from the human nervous system. It can be described as a structured machine learning protocol drawing parallels to the human brain. The protocol consists of layers featuring different processors (called neurons) tasked with critical processing information for the system. These neurons form connections with each other, thereby forming a network of neurons, and are capable of yielding acute outputs of a susceptible algorithm [20]. The system largely relies on these networks to solve a problem and utilize data set similarities and relationships. As a result, rather than being preprogrammed, the system develops through experience. It also increases mobility since data interpretation methods may be modified throughout the study. The network's main aim is to complete the specific tasks assigned to it and simultaneously learn from a benchmark provided, e.g., the system can be tasked to predict TN (true negative) and TP (true positive) values.

Additionally, an ANN can solve multi-dimensional issues and multi-task [21]-it creates a neural structure with processing units linked by coefficients (weights). With each iteration, the algorithm fluctuates the values of the importance and the biases such that the values generated by the algorithm are closer to the actual values. The neurons of the neural structure add up the product of weights and inputs to create a linear combination of the information delivered to an activation function. With back-propagation of mistakes during training iterations, the output is produced via feed-forward data flow. Above all, ANN models offer the advantage of forecasting the behavior of highly complex systems when traditional adsorption isotherms may be incorrect [22]. Owing to its capacity to simulate highly nonlinear systems, its application in complicated procedures such as multi-component adsorption has been effectively applied [23–27].

The Levenberg–Marquardt algorithm (LMA)-based ANN is recommended for nonlinear training because it combines gradient descent and quasi-Newton techniques to produce fast convergence and performance. Additionally, including LMA with second-order error optimization into ANN aids in overcoming the drawbacks of the back-propagation technique and delivers consistent and rapid results [28,29]. Hanandeh et al. (2021) [27] performed the adsorption modeling of Ni, Cu, and Pb ions using different types of solutions. This was achieved using biochar derived from date seed and depicted that LMA

was a fast algorithm, albeit not the best selection when the dataset in use is relatively more minor [30]. However, the study also featured another algorithm to compensate for this drawback—Bayesian regularization (BR), a slower algorithm but much more suitable for smaller datasets [27]. It is used to process a nonlinear regression as well-defined ridge regression, and the Bayesian models can be used to avoid overtraining and overfitting data. This is achieved by providing an objective Bayesian criterion for stopping training using evidence procedures [31].

It is proposed in this research to use human fecal ash as an adsorbent to adsorb urea from human urine, thus creating a self-sustaining sanitation loop. Wastes produced in this closed-loop system will be recycled and repurposed instead of being discharged into the environment, allowing us to manage resources more effectively while minimizing environmental externalities. The experimental results obtained are used to build an ANN model which is highly capable of predicting the recovery of urea from human urine. Different second-order methods are employed to train our network. The prediction accuracy and convergence time are then compared: Bayesian regularization techniques such as ANN, quasi-Newton, Levenberg–Marquardt, and conjugate gradient are considered, and the one that is the best match for our task is chosen.

The following objectives were assessed in our study: (i) the adsorption capacity of urea (sorbate) from synthetic urine was determined, (ii) the effect of different process parameters were evaluated, (iii) the data obtained were fitted with known isotherm models (Langmuir, Freundlich, Temkin) to understand the adsorption mechanism, (iv) an ANN model was developed using the data samples generated in our study, (v) possible end uses for fecal ash (sorbent) were identified and incorporated as a part of a closed nutrient loop.

## 2. Materials and Methods

### 2.1. Feedstock Preparation

The chemicals and reagents used for this investigation were obtained from SD Fine Chemicals, Mumbai, India. Human feces were produced in a controlled laboratory environment, according to the formulation given by the Pollution Research Group, 2014 (Table 1). Synthetic urine was prepared according to the suggestions and changes made by Ciba-Geigy (1977) [32], Pronk et al. (2006) [33], and Burns and Finlayson (1980) [34]. All raw ingredients for feedstock production were sourced from the local market in Vellore, Tamil Nadu, India, and were thoroughly mixed in the appropriate proportion to attain a homogeneous mixture. The mixture was placed in an airtight container and exposed to very high temperatures in an industrial-grade furnace $(T-14/HTF-1400-Technico, India)$ at 500 °C for 3 h (24 °C min$^{-1}$—heating rate). The resultant adsorbent, i.e., fecal ash was cooled down to room temperature, crushed, and sieved to achieve particle size in the range of 74 to 88 μm. Finally, the adsorbent was dried and kept in a desiccator until it was used.

**Table 1.** Composition of synthetic human feces prepared for the adsorption of nutrients from human urine.

| Ingredients | Dry Weight (g.kg$^{-1}$) |
|---|---|
| Baker's yeast | 72.8 |
| Peanut oil | 38.8 |
| Miso paste | 24.3 |
| Propylene glycol | 24.3 |
| Cellulose powder | 12.4 |
| Psyllium husk powder | 24.3 |
| Calcium phosphate | 25.0 |
| Water | 778.1 |

### 2.2. Batch Adsorption

Synthetic urine (100 mL) was mixed with different fecal ash masses in a series of experiments. The impact of mass loading of the fecal ash (0.5 g, 1.0 g, 1.5 g, and 2.0 g)

and effect of urea concentration (25%–3.375 g.L$^{-1}$, 50%–6.75 g.L$^{-1}$, 75%–10.125 g.L$^{-1}$, and 100%–13.5 g.L$^{-1}$) on the adsorption of essential plant nutrients was studied at different temperatures (30 °C, 35 °C, and 40 °C) for a total of 4 h. In total, 48 experiments were carried out in triplicate wherein one parameter varied while the rest were kept constant. All experiments were conducted using a thermostatic incubator shaker (Orbitek LT, Sciences Biotech, Chennai, India) at a speed of 150 rpm using 250 mL Erlenmeyer flasks. Aliquots of 5 mL were extracted, centrifuged (REMI Instruments, Mumbai, India), and passed through a Ministat filter (0.45 µm) at different time intervals. A Shimadzu UV mini–1240 spectrophotometer (Kyoto, Japan) (Peterson et al., 1961) [35] was used to measure the change in absorbance at $\lambda_{max}$ of 430 nm to estimate residual urea amounts. The absorbance was used to obtain the concentration of urea at time '$t$' ($C_t$). Taking the initial concentration ($C_i$) as 13.5 mg/L$^{-1}$, the amount of urea adsorbed under equilibrium conditions ($Q_e$) was also calculated for 3 different process parameters, i.e., temperature, mass loading, initial concentration of urine solution.

$$Q_e = (C_i - C_t) * \frac{V}{W}$$

here, $V$ represents the sorbate volume (L) and the amount of fecal ash is represented with $W$ (g).

### 2.3. Adsorption Equilibrium

Adsorption isotherms are mathematical models essential for understanding the adsorbate–adsorbent interactions when equilibrium is achieved. Equilibrium data obtained from the adsorbent were fitted against the adsorption isotherms [36,37]. Each model assumes different molecular theories that explain the governing mechanism of heavy metal ions onto fecal ash.

The Langmuir isotherm [38] is the most widely used isotherm, explaining the sorption of different compounds onto various sorbents. The maximal sorbate adsorption is expected to occur when a single layer of sorbate develops around the sorbent's surface, as represented by Equation (1)

$$\frac{1}{q_e} = \frac{1}{q_{max}} + \left[\frac{1}{b_{qmax}}\right]\left[\frac{1}{C_e}\right] \tag{1}$$

here, '$c_e$' (mg.L$^{-1}$) indicates the equilibrium concentration, '$q_e$' (mg.g$^{-1}$) denotes the total amount of urea adsorbed by the adsorbent under equilibrium circumstances, '$q_{max}$' (mg.g$^{-1}$) signifies the maximum equilibrium concentration, and '$b$' (L.g$^{-1}$) is just the equilibrium Langmuir constant. The preceding equation depicts a straight-line equation in which '1/$q_e$' versus '1/$c_e$' gives the slope '1/$b_{qmax}$' and the intercept '1/$q_{max}$'.

The Freundlich isotherm [39] is the empirical model for adsorption represented in a logarithmic form to provide a straight-line equation defined by Equation (2).

$$ln \ q_e = \ ln K_f + \frac{1}{n} \ ln \ C_e \tag{2}$$

here, '$K_f$' (mg.g$^{-1}$) and '$n$' represent the adsorption capacity of fecal ash and the associated adsorption intensity. Additionally, '$q_e$' (mg.g$^{-1}$) indicates the total urea adsorbed by the fecal ash at equilibrium, whereas '$C_e$' (mg.L$^{-1}$) specifies the concentration once equilibrium is reached.

Model fitting was used on the assumption that adsorption heat decreases linearly with increasing surface area. Thus, the Temkin model can actively correlate the binding and adsorption energies of the various chemical species present.

$$q_e = Bln(A) + Bln(C_e) \tag{3}$$

here in Equation (3), '$C_e$' (mg.L$^{-1}$) denotes the concentration at equilibrium, and '$q_e$' (mg.g$^{-1}$) is the total amount of urea adsorbed by the fecal ash at equilibrium. '$B$' (J.mol$^{-1}$) denotes the free energy associated with the sorbate–sorbent interaction, while '$A$' represents the adsorption capacity.

### 2.4. Development of ANN Model

The primary goal for this section of our study was to develop and test different ANN models and determine the model with the best fit that can be used to successfully predict the amount of urea adsorbed. To cater to this purpose, MATLAB R2015b was employed (consisting of the Toolbox of neural network) to create the ANN models. The networks are composed of three layers: an input layer with three nodes, i.e., the 3 process parameters used in this study (temperature, initial concentration, and mass loading), an output layer with one node yielding the final result (adsorbed urea), and a hidden layer with fourteen neurons. Three-layered ANN models were constructed and evaluated to determine the most accurate prediction models for this research. Tangent sigmoid functions are used as transfer functions for the created ANN models' hidden and output layers. The impact of varying the number of neurons in the hidden layer on the final mean square error was analyzed, and the optimal number was chosen. Numerous systems based on various artificial neural network models were created using data collected during a series of experiments. The Levenberg–Marquardt algorithm (TRAINLM), scaled conjugate gradient, Bayesian regularization (TRAINBR), and quasi-Newton were among the back-propagation techniques evaluated to get the optimum fit.

The development of the ANN models was based on the dataset obtained from the experimental results of our study. The dataset mentioned above was subdivided into 3 subgroups—training (70% of data), validation (15% of data), and testing (15% of data), followed by a subsequent normalization of these subgroups as per Equation (4).

$$x_{n,i} = \frac{x_i - data_{min}}{(data_{max} - data_{min})} * (R_{n,upper} - R_{n,lower}) + R_{n,lower} \tag{4}$$

Here, '$x_{n,i}$' represents the normalized value of '$x_i$', '$data_{min}$' denotes the minimum value of data, '$data_{max}$' represents the maximum value of data, '$R_{n,upper}$' represents the normalized upper range, and '$R_{n,lower}$' denotes the normalized lower range. Subdivision of the dataset was performed to negate instances of overtraining the models and, to avoid any computational issues occurring in the ANN models, data normalization was adopted. The epoch was initially set at 1000. Multiple linear regression analysis (MLR) was performed for our dataset to predict the TN and TP. Numerous evaluation techniques were used to comprehend, verify, and evaluate the models' fit and prediction accuracy, including root mean square error, absolute average deviation, R-squared, and adjusted R-squared. These techniques compared the models' predicted values to the observed values of TN and TP. Lastly, the autocorrelation of error with a variation of lag was calculated and minimized for each model, and the mean squared error was calculated as shown in Equation (5).

$$MSE = \frac{1}{n} \sum_{i=1}^{n} (Y_i - \hat{Y}_i)^2 \tag{5}$$

## 3. Results and Discussion

### 3.1. Adsorption Kinetics

The mass loading of fecal ash and the initial urine concentration are two critical parameters determining the maximum nutrient removal efficiency during adsorption experiments. As a result, it is essential to predict the range of values for the above mentioned factors, which would maximize nutrient recovery efficiency. As shown in Figure 1a, with fecal ash loading of 0.5 g–1.5 g, urea sorption was high, exceeding 660.37 mg.g$^{-1}$; this may be due to the existence of sorption sites on the adsorbent's surface. When the dose was raised from 1.5 to 2.0 g, a significant decrease in equilibrium adsorption capacity occurred (results decreased to 481.73 mg.g$^{-1}$). This was mainly due to the adsorbent material's abundance of sorption sites compared to the solution's urea concentration. As the mass loading varied between 0.5 and 2.0 g, urea removal increased from 80.63% to 92.80%. One



gram of loading was shown to be the optimal dosage for 0.1 L of synthetic urine, with an increase in urea recovery of less than 5% for loading doses of 1.0–1.5 g.

A set of experiments were performed at various urine concentrations to see how the initial urine concentration affects mass transfer resistance between the adsorbent and adsorbate. To achieve different initial concentrations, deionized water diluted the urine (from its original concentration), thereby achieving varying initial concentrations. Keeping the adsorbent loading of fecal ash at a constant 1.0 g, the effect of varying initial concentrations of urine (between 25% and 100%) was studied. It was observed that when the urea concentration was increased from 25% to 100%, the adsorption capacity increased from 547.42 to 1242.13 mg.g$^{-1}$. The likelihood of urea and fecal ash molecules colliding rises with concentration, allowing urea molecules to overcome the barrier seen between the solid and aqueous phases. The smooth and continuous curves obtained at all concentrations indicate that monolayer urea may be adsorbed onto the ash [24,40]. It took 180 min for the sorption to achieve equilibrium. Figure 1b, demonstrates that a significant quantity of urea was adsorbed during the first 120 min, following which stirring was used to reach equilibrium with very little urea being adsorbed. The large surface area of the fecal ash accounts for the initial phase's high adsorption capability. The increased surface area over the fecal ash is the increased uptake capacity during the initial phase. Until 120 min, a thick one molecular layer of urea is produced over the fecal ash, at which point the rate of urea transfer regulates the rate of sorption from the outside to the inside of the ash. Similar findings have been obtained in many investigations employing activated carbon to remove carboxylic acids, dyes and heavy metals [41–43]. Urea recovery was higher than 80% at all concentrations, implying that the starting concentration has no impact on the quantity of urea recovered at equilibrium.

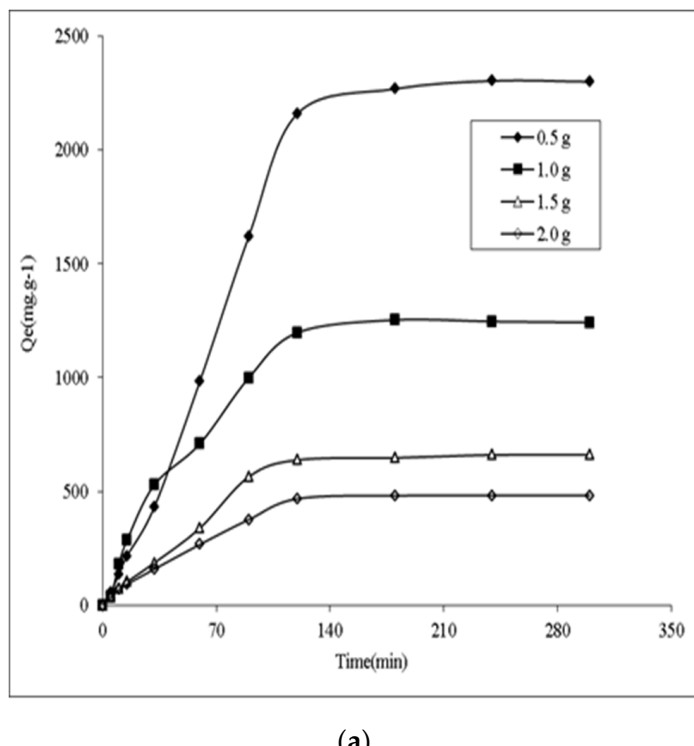

**(a)**

**Figure 1.** *Cont.*

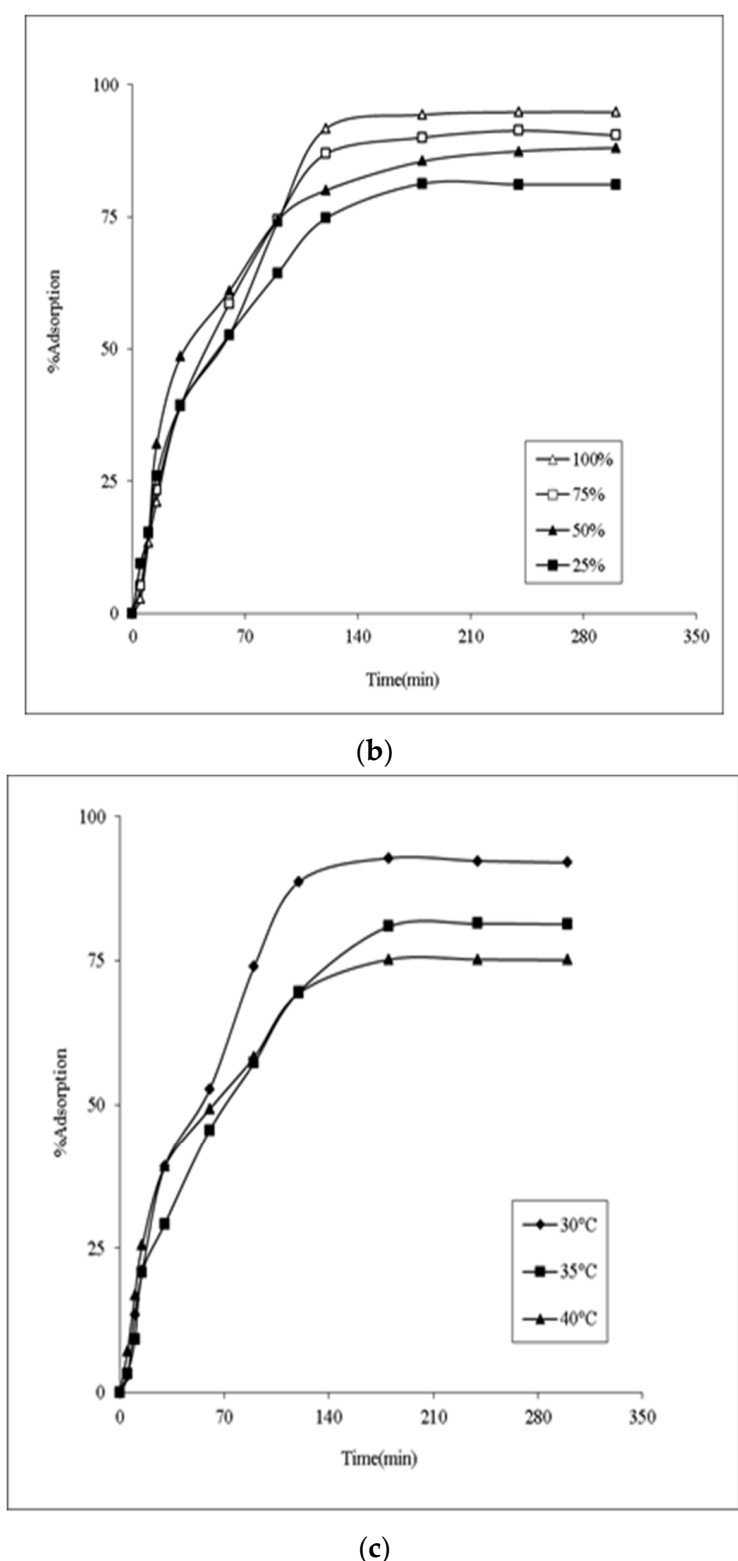

**(b)**

**(c)**

**Figure 1.** The effect of process parameters on adsorption of urea. (**a**) Fecal ash loading (initial urine concentration—13.5 g.L$^{-1}$, temperature—30 °C); (**b**) initial urine concentration (fecal ash loading—1.0 g, temperature—30 °C); (**c**) temperature (fecal ash loading—1.0 g, initial urine concentration—13.5 g.L$^{-1}$).

The impact of temperature was investigated using a shaker speed set to a constant 150 rpm and a fixed adsorbate concentration of 13.375 g.L$^{-1}$, and a fixed adsorbent loading of 1.0 g. The experiments were carried out at 3 unique incubator temperatures, and

the adsorption capacity was found to decrease from 92.8% at 30 °C to 75.13% at 40 °C (Figure 1c). As a result, the urea adsorption capacity was 1242.13 mg.g$^{-1}$, 1097.55 mg.g$^{-1}$, and 1013.85 mg.g$^{-1}$ at 30 °C, 35 °C, and 40 °C, respectively, consistent with an exothermic nature of adsorption. This demonstrates that exothermic adsorption occurs to remove urea from fecal ash. Under low-temperature circumstances, the average molecular energy is smaller, which actively enhances the interaction of urea with fecal ash, promoting its adsorption. Gupta et al. (2001) [44] reported a similar trend in pesticide removal, which ascribed to the exothermic nature of organic chemical sorption. Considering all concentrations and setting the reference sorption temperature as 30 °C, 11.67% was the average reduction of urea sorption for 35 °C and 18.37% for 40 °C. The entropy of sorption rises as the kinetic energy of the adsorbate molecules increases as the system temperature increases. This has a detrimental effect on the adsorbate molecules' capacity to collect on the surface of the fecal ash adsorbent. This is consistent with the research on activated carbon as an adsorbent. Dąbrowski et al. (2005) [45], Chiang et al. (2001) [46], and Babel et al. (2003) [47] all reached similar results regarding the adsorption of substances including phenolic, volatile organic compounds, and heavy metals.

### 3.2. Isotherm Models

The adsorption phenomenon depicted an escalation for the initial 120 min after the commencement of the experiment. This was followed by a slow-down phase between 150 and 180 min as the system reached its equilibrium. The best-fit parameters for each adsorption model are listed in Table 2. While comparing with Langmuir and Freundlich, it was seen that the best results were obtained in the Temkin model, suggesting that the adsorption of urea from wastewater followed the homogeneous or heterogeneous processes.

**Table 2.** Analysis of different types of isotherm models with two parameters for recovery of urea.

| Isotherms | Parameters | Fecal Ash | | | | | | | |
|---|---|---|---|---|---|---|---|---|---|
| | | Initial Concentration (g.L$^{-1}$) | | | | Mass Loading (g) | | | |
| | | 3.375 | 6.75 | 10.125 | 13.5 | 0.5 | 1.0 | 1.5 | 2.0 |
| Langmuir | $R^2$ | 0.184 | 0.259 | 0.359 | 0.464 | 0.295 | 0.184 | 0.440 | 0.475 |
| | $q_m$ (mg. g$^{-1}$) | 125 | 136.9 | 116.2 | 100 | 131.5 | 125 | 52.91 | 49.01 |
| | $K_L$ (L.mg$^{-1}$) | 0.953 | 1.177 | 1.228 | 0.847 | 0.463 | 0.952 | 0.273 | 0.265 |
| Freundlich | $R^2$ | 0.656 | 0.721 | 0.706 | 0.825 | 0.767 | 0.655 | 0.845 | 0.849 |
| | N | 1.172 | 1.219 | 1.511 | 0.917 | 0.692 | 1.171 | 0.559 | 0.580 |
| | $K_f$ (mg. g$^{-1}$) (g$^{-1}$)$^n$ | 1619 | 1029 | 637.5 | 843.1 | 8228 | 1619 | 7767 | 5748 |
| Temkin | $R^2$ | 0.929 | 0.932 | 0.887 | 0.913 | 0.967 | 0.929 | 0.986 | 0.984 |
| | B (J.mol$^{-1}$) | 42702 | 332 | 190.2 | 291.3 | 1153 | 427.2 | 474 | 359 |
| | A | 22.42 | 16.67 | 22.57 | 9.059 | 16.23 | 22.42 | 15.15 | 15.34 |

Adsorption isotherms further revealed that the fecal ash exhibits a comparatively higher sorption capacity that can be estimated to be greater than 481.73 mg.g$^{-1}$ for urea present in the aqueous solutions. The $R^2$ values for the linearized isothermal adsorption models were less than 0.46 for Langmuir and higher than 0.7679 and 0.9672 for Freundlich and Temkin, respectively, suggesting that the Temkin isotherm model matched the equilibrium adsorption data better (Table 2). This further ascertains those multiple mechanisms could regulate the adsorption of urea onto fecal ash.

### 3.3. ANN Model

The experimental results obtained from our adsorption studies were used to build and train various ANN models to assess urea recovery from human urine using fecal ash. Different second-order methods were employed to train our network. The prediction accuracy and convergence time were then compared: Bayesian regularization techniques such as ANN, quasi-Newton, Levenberg–Marquardt, and conjugate gradient were considered, and the one that was the best match for our task was chosen. Out of the selected models, only two (TRAINBR and TRAINLM) exhibited considerable output performance for our study. Hence, the other models were disregarded as their performance was marginal to fit in our study. The datasets used in this study were segmented and normalized as input data for the various ANN models. This proved beneficial as none of the models in our study exhibited instances of overtraining and faced minimal computation-related problems while training the ANN models.

The study results indicated that incorporating neurons in the hidden layer led to an improvement in performance and the efficiency of prediction. The actual values TN and TP and the predicted values obtained from MLR were compared to validate and assess the prediction accuracy of each model and compute the goodness of fit for each of them. The error values were calculated with absolute average deviation (AAD), adjusted-$R^2$, R-squared ($R^2$), and root mean square error (RMSE). The residual errors obtained from the abovementioned methods were closer to zero for two models (TRAINLM and TRAINBR). This demonstrates favorable compatibility among the actual values and the models' outputs. In general, for a model to be deemed as the best model, it must exude minimal error values and highest counts in terms of correlation coefficients. Similarly, in our study, the best structure of the ANN model was decided based on the minimal error approach. Hence, BRANN was considered as a superior ANN model in this study.

Cross-validation is no longer necessary with Bayesian regularized artificial neural networks (BRANNs) (Figure 2a). This aided in the validation set selection and network design optimization processes. Additionally, BRANNs may be utilized with automatic relevance determination (ARD) of the input variables, allowing the network to "guess" the significance of each input further. This avoids the inclusion of unnecessary or highly correlated indices in the modeling process and identifies the most critical factors for generating the activity data. Training performance using optimization is executed for 1000 epochs. Based on the mean squared error, the best training performance is achieved at 144 epochs, with an optimized value of 1.0077 for Bayesian regularization (Figure 2b). Obtaining the variations using the gradient method based on the gain leads to the error being minimized. Step by step, we can analyze the rate of change, thereby regularizing the parameters (Figure 2c). Error histograms with 20 bins were used (Figure 2d). It was seen that by using the Bayesian network, the error value could be reduced to zero. The autocorrelation of error with variations of lag leads to a minimized error with the confidence limits, as shown in Figure 2g. Responses of the outputs using Bayesian regularization are recorded with zero error, and accuracy of R = 0.9982 was recorded with a convergence time of 7 s (Figure 2e,f).

The Levenberg–Marquardt algorithm, also referred to as the damped least-squares technique, was developed especially for loss functions that take the form of a sum of squared errors. It is valid even if the precise Hessian matrix is not computed—instead of that, it utilizes the gradient vector and the Jacobian matrix. A neural network is designed to analyze the performance of mean squared error using Levenberg–Marquardt (Figure 3a). For the given model, the best validation performance is obtained at epoch 1, and the value is 31.52 (Figure 3b). Obtaining the variations using the gradient method based on the gain leads to the error being minimized. Step by step, we can analyze the rate of change, thereby regularizing the parameters. Still, unlike the Bayesian network, it fails at epoch 5 (Figure 3c). The deviations between targets and outputs are obtained such that the error is minimized. Twenty bins are taken to visualize the error using a histogram as shown in Figure 3d. Variation of lag using autocorrelation of error is calculated, and the confidence limits are

obtained as done for Bayesian (Figure 3g). Response of the output element was generated for the Levenberg model, and it was seen that the 'R' value was found to be 0.9941, and the fastest convergence time was noted at 7.2 s (Figure 3e,f). These findings demonstrate that ANNs are effective and reliable methods for analyzing the efficacy of fecal ash as an adsorbent for urea recovery from human urine. A comparative analysis of all optimization algorithms revealed that the Bayesian regularization algorithm is capable of displaying potentially complex relationships and has the best performance when used in conjunction with other methods to determine the feasibility of using fecal ash as a prospective adsorbent for the recovery of plant essential nutrients from wastewater on a larger scale.

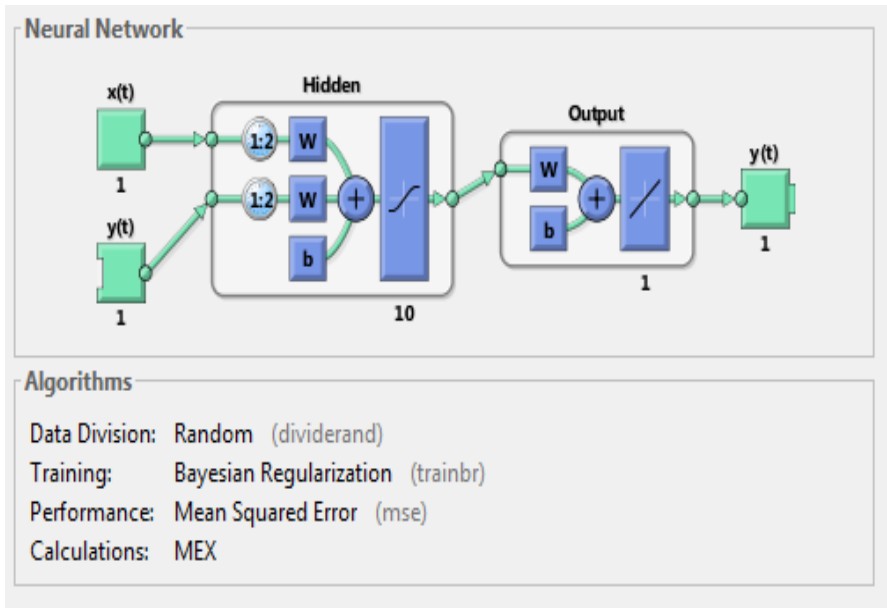

(**a**)

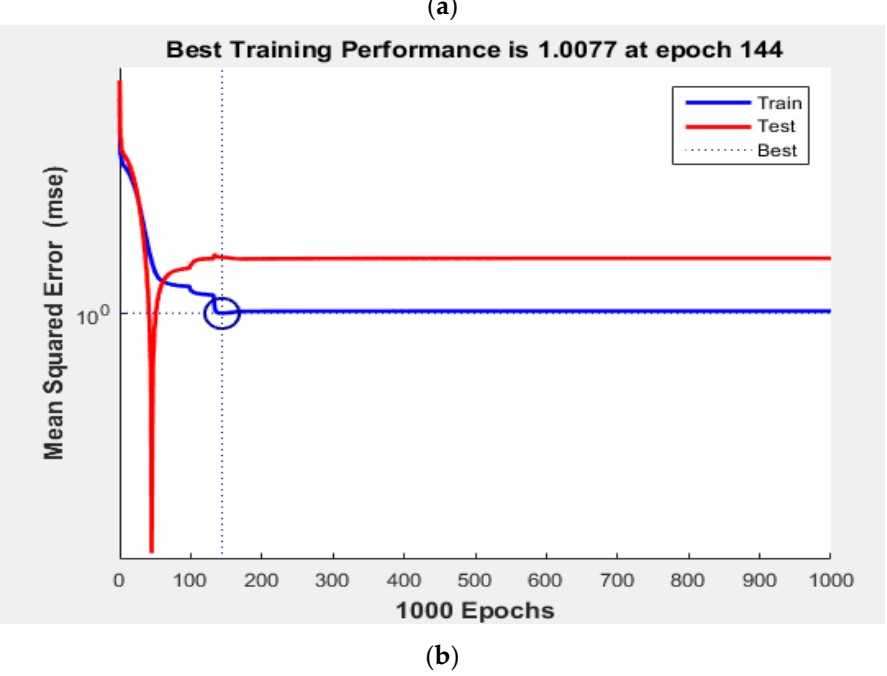

(**b**)

**Figure 2.** *Cont.*

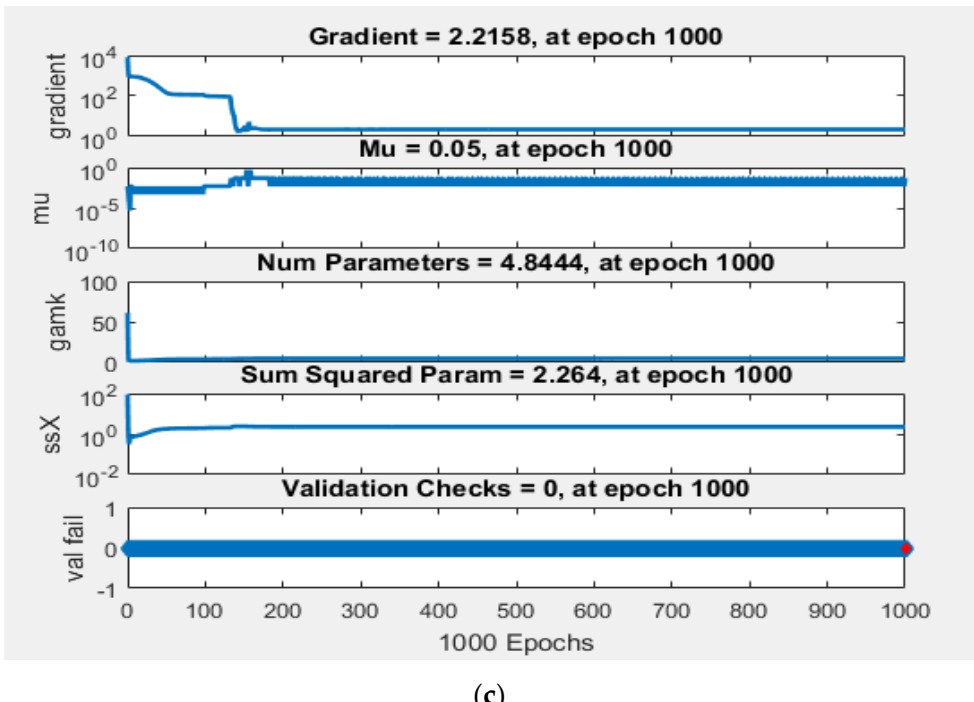

(**c**)

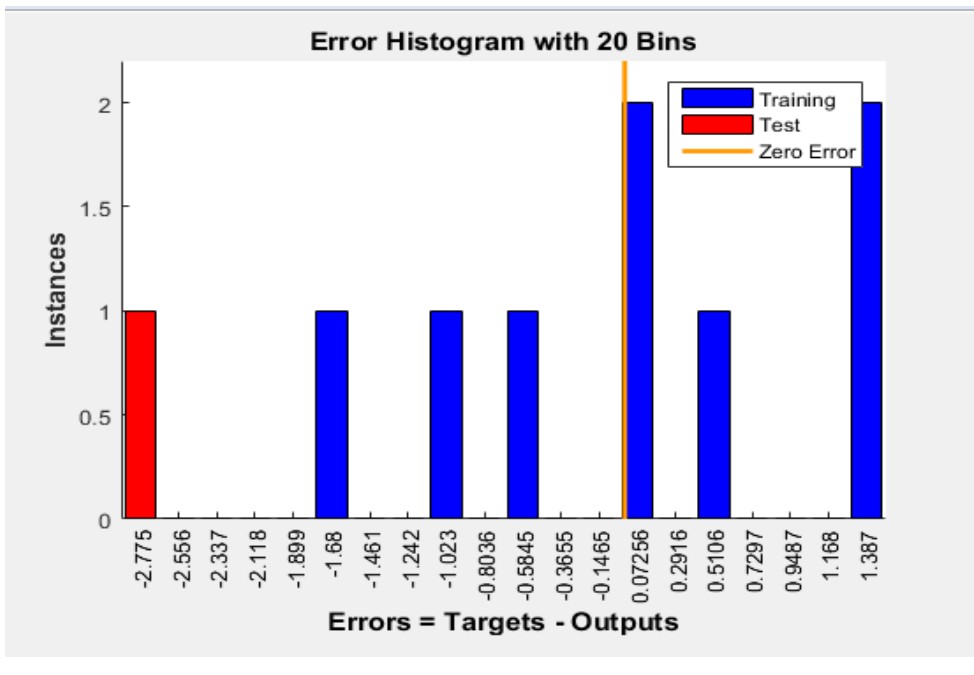

(**d**)

**Figure 2.** *Cont.*

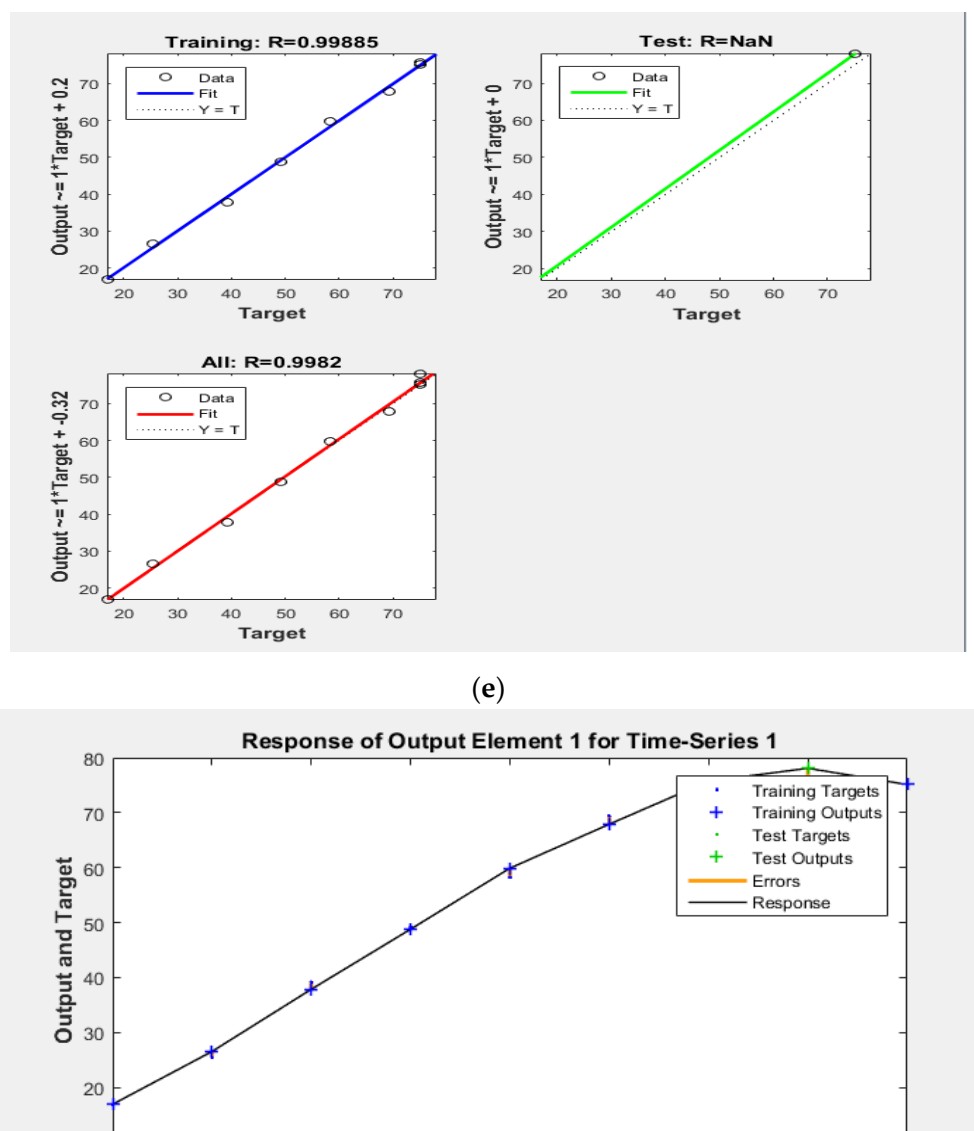

(**e**)

(**f**)

**Figure 2.** *Cont*.

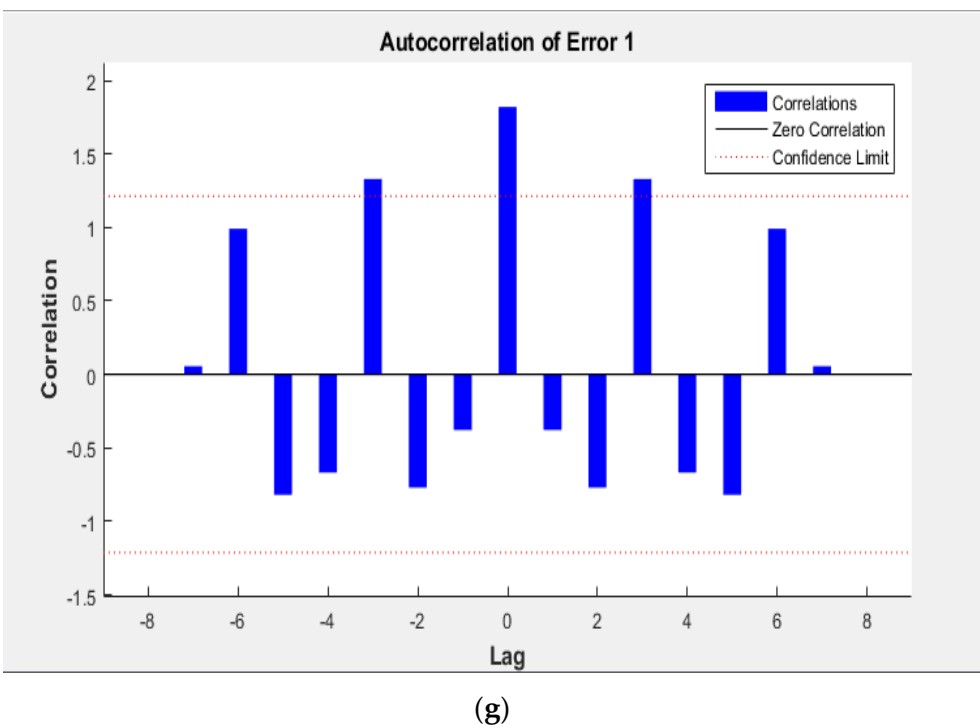

(**g**)

**Figure 2.** Bayesian regularization artificial neural network (BRANN) for urea adsorption. (**a**) Neural network overview; (**b**) training performance; (**c**) variation in gradient, training gain (mu) and validation fail for epoch range 0 to 5; (**d**) error histograms with 20 bins; (**e**) response of output element; (**f**) results of curve fitting with variation in target; (**g**) autocorrelation of error with variation of lag.

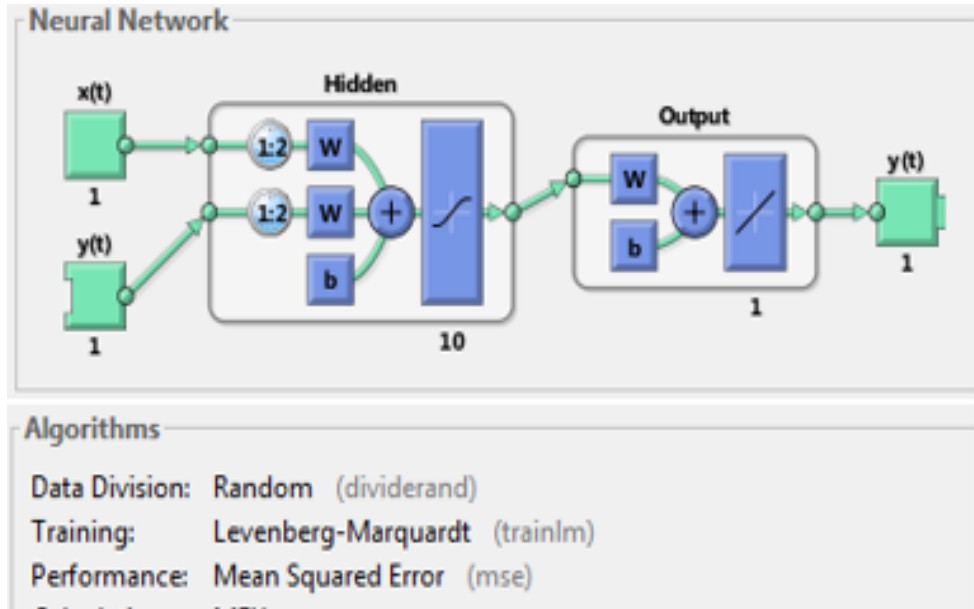

(**a**)

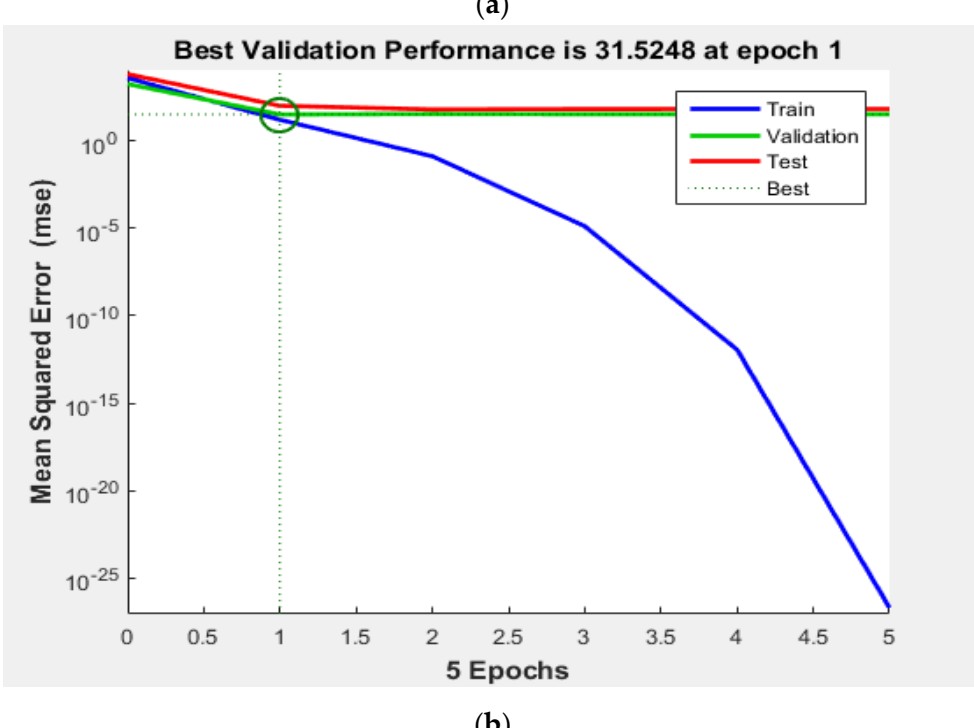

(**b**)

**Figure 3.** *Cont.*

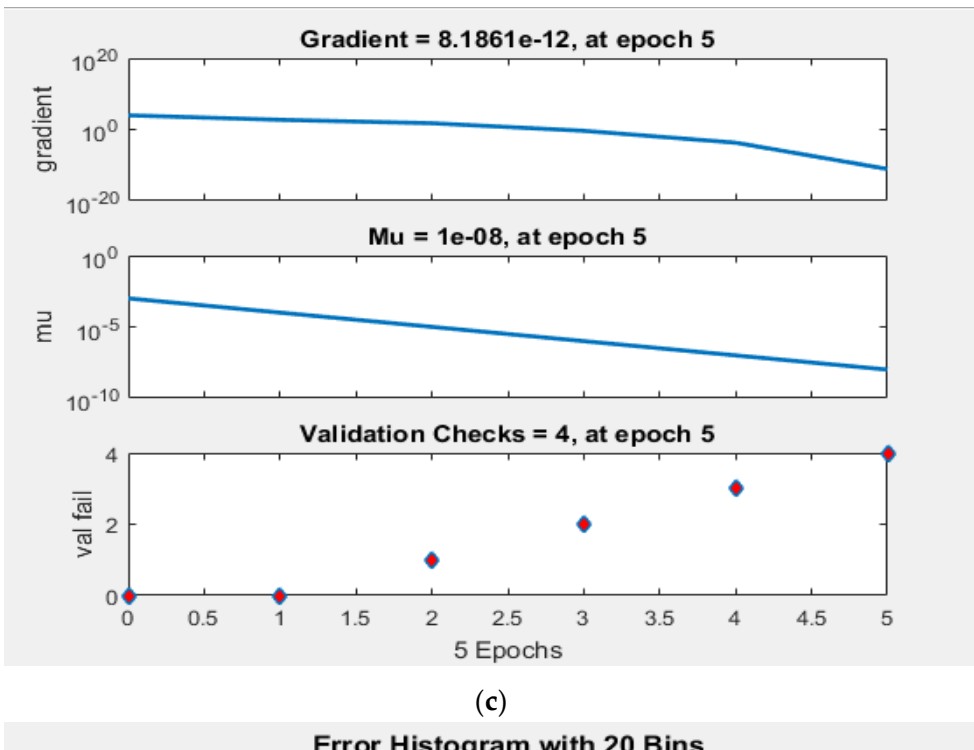

(**c**)

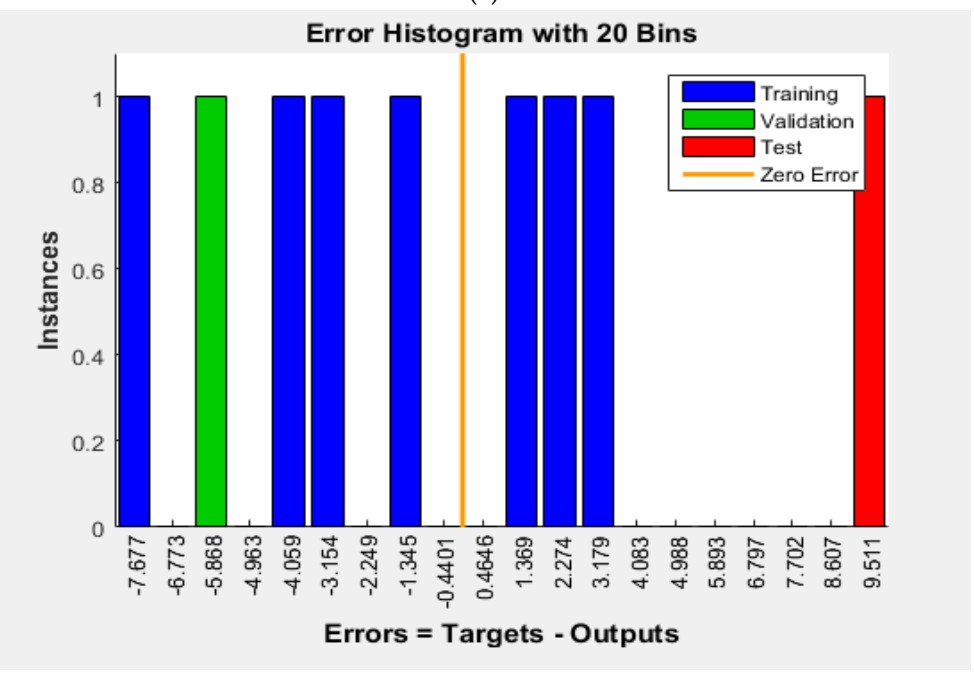

(**d**)

**Figure 3.** *Cont.*

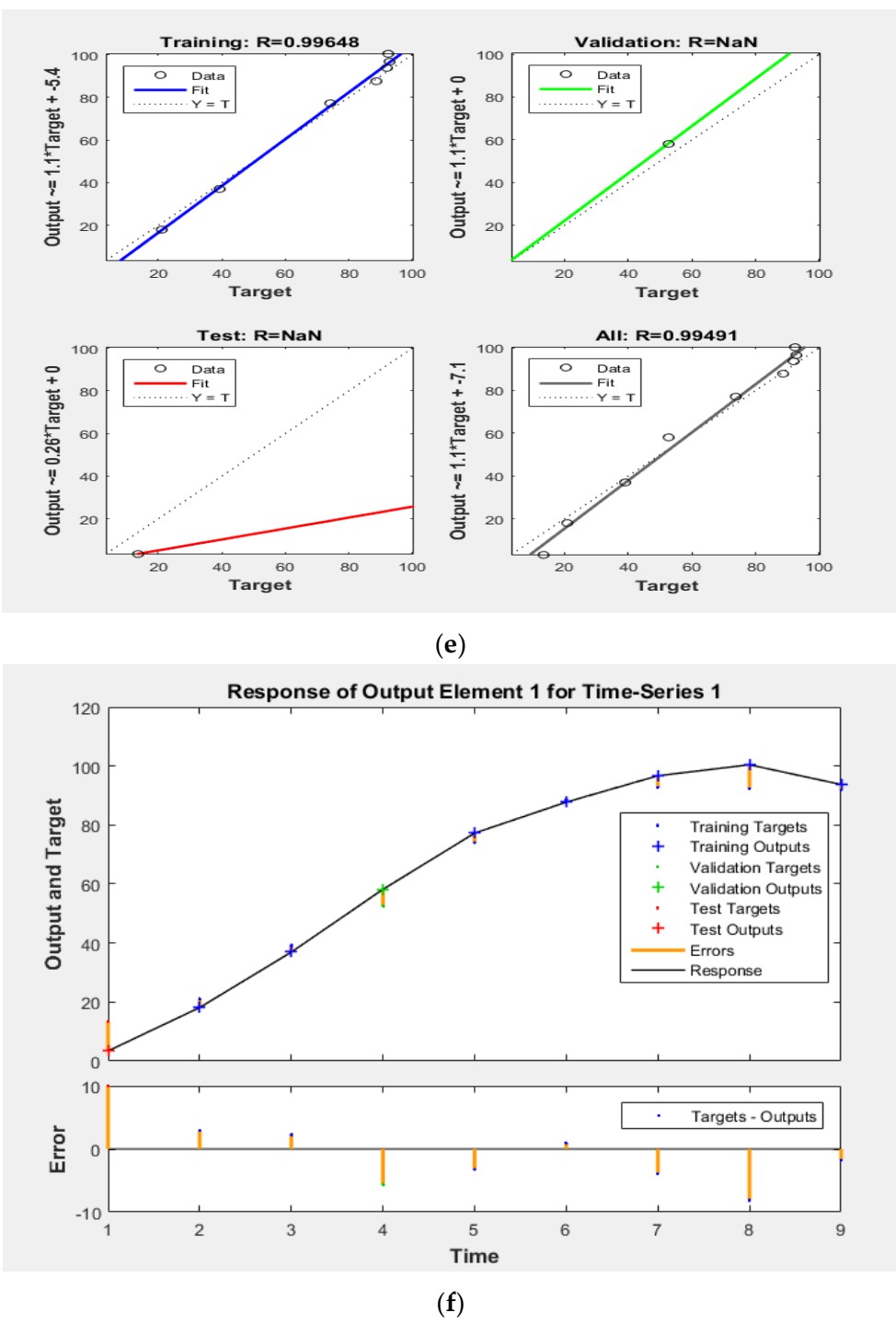

(**e**)

(**f**)

**Figure 3.** *Cont.*

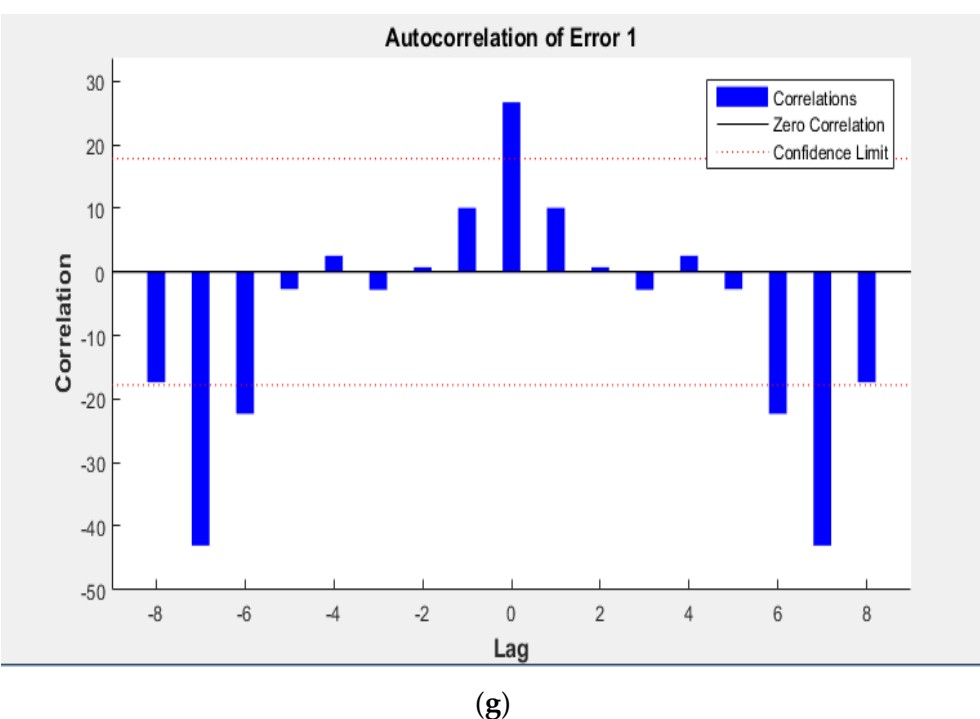

(**g**)

**Figure 3.** Levenberg–Marquardt back-propagation neural network (LMBNN) for urea adsorption. (**a**) Neural network overview; (**b**) training performance; (**c**) variation in gradient, training gain (mu) and validation fail for epoch range 0 to 5; (**d**) error histograms with 20 bins; (**e**) response of output element; (**f**) results of curve fitting with variation in target; (**g**) autocorrelation of error with variation of lag.

## 4. Applications and Societal Importance

Fecal ash derived from the treatment of human wastes has a minimal environmental impact, and it could be an effective solution for the management of solid wastes. Simultaneously, studies reveal that fecal ash can also serve varied applications in treatment plants and as a consumer-grade product. Primarily, fecal ash can be used as an excellent adsorbent material owing to its tendency to trap moisture and other types of soil nutrients. Although the adsorption capacity is not on par with other carbonaceous materials such as activated carbon—fecal ash compensates for this lack in performance by being a "green" adsorbent. Pinatha et al., 2020 [48], conducted a study to recover phosphorous from human urine using solid waste ash. The study deemed fecal ash as an effective solution to adsorb phosphorous from wastewater streams.

Fecal ash can also be used for soil remediation—used ash contains moisture and soil nutrients embedded in it. This can be added to plantation fields as a fertilizer and will serve as a viable method of closing nutrient loops in an ecosystem. Hence, it can be a cost-effective and eco-friendly alternative to chemical fertilizers. Other benefits may include a considerable reduction in solid waste disposal cost, which can significantly lower the operational cost of a sewage plant. Produced ash can be valorized and serve tertiary applications instead of ending up in landfills.

Despite all the potential applications of fecal ash, this ubiquitous resource continues to be flushed out every day. Aside from societal perception, this oversight practice can be primarily accredited to the lack of reliable data that can be used to quantify and assess this resource as a commercial-grade product. Hence, there is an acute necessity to set up a uniform database of information that producers and traders can refer to. Lastly, there is a shortage in literature for comprehensive multi-variable optimization studies that can accurately assess the produced ash by taking numerous process parameters and variables

into consideration. Such a study will also help predict and classify the ash produced according to their most suitable application.

The lack of proper sanitation facilities is a prevalent issue in several countries to date. Although most nations are actively ensuring appropriate sanitation facilities, the solid waste collected from these establishments is not adequately managed. This study plays its part by offering a viable eco-friendly solution to the question of solid waste management

Simultaneously, proper treatment of such biowastes mitigates the spread of diseases and allows an increased number of sanitation facilities. This also uplifts the living standards of the general population and puts a stop to the practice of open defecation. Hence, the study lays a foundation to rehabilitate and remediate different sectors of our society.

## 5. Conclusions

The trials conducted in our study revealed that fecal ash could serve as a practical solution for the sorption of urea. Maximum urea adsorption was recorded at 92.8% under an initial concentration of 13.5 g.L$^{-1}$ with an adsorbent loading of 1.0 g at a low temperature of 30 °C. The results were also validated by both ANN models, i.e., TRAINLM and TRAINBR yielding identical optimized parameters for the urea adsorption. The Temkin model depicted the best results in terms of coefficient of correlation, indicating a tendency of the adsorption phenomenon to follow homogenous or heterogeneous processes. The results from our study further illustrate the effectiveness of ANN models for reliably analyzing the efficacy of fecal ash as an adsorbent for urea recovery from human urine. Bayesian regularization algorithm exhibited the highest accuracy of prediction for revealing potentially complex relationships and to determine the feasibility of using fecal ash as a prospective adsorbent for the recovery of plant essential nutrients (such as urea from wastewater) on a larger scale. Although fecal ash's adsorption efficiency falls short of conventional carbon-based adsorbents (e.g., activated charcoal), it compensates for this shortcoming by being a practical, sustainable solution. Most of these carbon-based adsorbents accumulate over soil or have to be replenished after the process of adsorption.

On the contrary, fecal ash can be used as an alternative for fertilizers after the adsorption process. This enables urea adsorbed from the wastewater stream to be directly supplied back into agricultural soil and leads to a simultaneous valorization of human waste. Lastly, it also provides an adequate answer for wastewater treatment, thereby providing a self-sustainable solution to sanitation by treating human waste with human waste.

**Author Contributions:** Design of experiments, M.G. and J.D.; experimentation and data collection, A.T., A.S. and R.M.; analysis of the experiments, M.G.; ann modeling, V.C.; manuscript draft, A.T., A.S. and R.M.; editing and reviewing, A.S., M.G., J.D. and V.C. All authors have read and agreed to the published version of the manuscript.

**Funding:** The authors declare that no funds, grants, or other support were received during the preparation of this manuscript.

**Data Availability Statement:** Not applicable.

**Conflicts of Interest:** All authors certify that they have no affiliations with or involvement in any organization or entity with any financial interest or non-financial interest in the subject matter or materials discussed in this manuscript.

## Nomenclature

| | |
|---|---|
| $A$ | Adsorption Capacity |
| $b$ | Equilibrium Langmuir Constant (L.g$^{-1}$) |
| $B$ | Free Energy Associated with the Sorbate–Sorbent Interaction (J.mol$^{-1}$) |
| $C_e$ | Equilibrium Concentration of Urea (mg.L$^{-1}$) |
| $C_i$ | Initial Concentration of Urea (mg.L$^{-1}$) |
| $C_t$ | Concentration of Urea at Time t (mg.L$^{-1}$) |
| $data_{max}$ | Maximum Value of Data |
| $data_{min}$ | Minimum Value of Data |
| $K_f$ | Adsorption Capacity of Fecal Ash (mg.g$^{-1}$) |
| $n$ | Adsorption Intensity |
| $q_e$ | Amount of Urea Adsorbed under Equilibrium (mg.g$^{-1}$) |
| $q_{max}$ | Maximum Equilibrium Concentration of Urea (mg.g$^{-1}$) |
| $R_{n,lower}$ | Normalized Lower Range |
| $R_{n,upper}$ | Normalized Upper Range |
| $V$ | Sorbate Volume (L) |
| $W$ | Amount of Fecal Ash (g) |
| $x_i$ | ith Data Point |
| $x_{n,i}$ | Normalized Value of $x_i$ |
| $\Lambda_{max}$ | Maximum Wavelength (nm) |
| AAD | Absolute Average Deviation |
| ANN | Artificial Neural Network |
| ARD | Automatic Relevance Determination |
| BR | Bayesian Regularization |
| BRANNs | Bayesian Regularized Artificial Neural Networks |
| LMA | Levenberg–Marquardt Algorithm |
| MLR | Multiple Linear Regression Analysis |
| MSE | Mean Square Error |
| RMSE | Root Mean Square Error |
| TN | True Negative |
| TP | True Positive |

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
