# Peer review of "Design and Analysis of Artificial Neural Network (ANN) Models for Achieving Self-Sustainability in Sanitation"

_applsci, doi:10.3390/app12073384_

Round 1
Reviewer 1 Report
The paper sounds good from the technical side. However, some minor corrections are still required.
To improve readability, a list of abbreviations and symbols must be used/added in the manuscript.
In the literature review, part is lacking, the author should add more latest research findings and results here.
The abstract and the conclusion sections can be improved or rewritten, especially the conclusion section as they are not precisely explaining the ending.
Please correct the grammar throughout the manuscript.
Reviewer 2 Report
The manuscript entitled ‘A step towards attaining self-sustainability in sanitation – ANN approach’ deals with using fecal ash as urea adsorbent. The experimental results were applied to build an ANN model. In my opinion, the results are positive with interesting inputs and outputs, however, some concerns need to be addressed to improve the quality of the work. Thus I can recommend major revisions to enable this manuscript to be published anywhere. Specific comments are:
- ‘introduction’, line 22-23: ‘and the acquired data were fitted to isotherm models to determine the adsorption kinetics.’ Do the authors know the difference between isotherm models and kinetic models? Isotherm models are NOT used to describe kinetics.
- ‘Keywords’: please add what is sorbate and sorbent
- line 142: ‘They were thoroughly mixed… ‘ what was mixed?
- Line 146: ‘crushed, and sieved to achieve uniform particle size distribution.’: What sieve? What is the particle size distribution?
- ‘Batch adsorption’ section, line 168: please add the literature on which the spectrophotometric determination of urea was based.
- ‘Batch adsorption’ section: please insert the formulas which you used to calculate the concentration of urea in the aqueous and solid phase
- ‘Adsorption Equilibrium’ section, line 178: ‘mechanism of heavy metal ions onto fecal ash.’? Heavy metals?
- line 184-188: please fill in the missing units when describing the symbols.
- section ‘Effect of Process Parameters’: please choose to use ‘fecal ash’ or ‘fecal char’ and use this option throughout the article.
- Figure captions are usually given below the figure and not above
- Section ‘Adsorption Kinetics’: ’ Do the authors know the difference between isotherm models and kinetic models? Langmuir, Freundlich and Temkin models do not describe kinetics.
- line 361- ‘The fecal ash in aqueous solutions exhibits comparatively faster adsorption kinetics’ this sentence is not understandable
- line 369 and 374: this manuscript is related to nitrogen or urea adsorption?
- The authors should consider improving the title of the article and include the use of ANN to predict the sorption process
Round 2
Reviewer 2 Report
accepted